# OpenReview forum: "AutoRT: Embodied Foundation Models for Large Scale Orchestration of Robotic Agents"
_ICLR.cc/2024/Conference — Submitted to ICLR 2024_

### Official Review · Reviewer_mKqw · 2023-10-30

**Soundness:** 3 good
**Presentation:** 3 good
**Contribution:** 2 fair
**Rating:** 5
**Confidence:** 4

**Summary:**

This paper introduces AutoRT which is a system designed to address the challenge of training autonomous robots in real-world environments with minimal human supervision. The main components consist of two parts: 1) VLMs discover environmental affordances from visual observations. 2) LLM proposes the tasks to try based on prompts and the affordances from VLMs. Basically, AutoRT leverages LLMs and VLMs to enhance the autonomy and data collection capabilities of a fleet of robots. The system employs multiple "collect policies" to ensure diverse data collection and adaptability, and tasks are filtered based on the capabilities of the selected policy. Finally, the authors extensively evaluate AutoRT in various real-world scenarios resulting in the collection of 77,000 trajectories.

**Strengths:**

- The idea of autonomous embodied agents with foundation models from collecting data to exploring the world is interesting and the proposed system is quite well-organized for that.
- This paper is well-written and easy to follow. And the method is clearly presented with descriptive figures.
- This paper shows extensive real-world experiment evaluations in diverse tasks and  environments. And the results demonstrate that the data collected by AutoRT improves RT-1’s performance, which shows that autonomous embodied system can collect useful data for action learning and this line of work is the promising way to go in the future.

**Weaknesses:**

- This paper lacks of technical novelty. This is a simple extension of [1, 2] to real-world robotic scenarios. I wonder if there is unique technical contribution over these existing works to deal with real-world robotic scenarios.
- In table 1, the collect policies other than teleoperation have very low success rate which means the collected data might not be useful for action learning. Basically, this method sounds like it is scalable but in a low quality. Can we learn meaningful actions from these data other than visual generalization ability?
- In Figure 3, it seems that teleoperated data has diverse action policies. However, when looking at the other data, excluding teleoperated data, they do not seem to have diverse action policies, especially considering that there are over 20 times more episodes with scripted policies. It seems that AutoRT does not explore sufficiently. Is there a way to encourage more action diversity?


[1] Yuqing Du et al., Guiding Pretraining in Reinforcement Learning with Large Language Models. ICML, 2023.\
[2] Guanzhi Wang et al., VOYAGER: An Open-Ended Embodied Agent with Large Language Models. arXiv, 2023.

**Questions:**

- Does LLM ever propose tasks that require skills not within the robot's scripted policy or teleoperation policy? If so, how does the robot behave in such cases? If such cases do not exist, how does the LLM restrict task proposals to the skills the robot possesses?
- I'm curious about how robots automatically perform a reset after completing a task. Is there a predefined reset policy, or is the reset also part of the task itself? I'm curious because there seems to be not much information on the reset process in the paper.

---

> ### Author Response · Authors · 2023-11-17
> **Rebuttal reponse to reviewer mKqw part 1**
>
> Thank you for taking the time to review the paper. AutoRT aims to be a data collection system that is agnostic to the specifics of collection and learned policies such that it can deploy a wide variety of collection policies to mine data that may be useful for multiple learnable policies. As such, the focus of this paper is not on designing new collect or learned policies, but to develop a framework that orchestrates a number of potential collection policies (automated or teleoperated) to increase the diversity of the collected data -- which in turn can enhance policy learning.
>
> A main concern of the reviewer is whether there is technical novelty given prior methods for simulated tasks. The real world is significantly different from sim. In real it is harder to scale, act autonomously, act safely, or know the ground truth state of the world, and there is technical novelty in how to tackle these real-world specific challenges, which we discuss below. Additionally, we’d like to note that there is significant interest in the robotics community for improving real world data collection. RT-2, a model using Internet-scale image data, still fails to generalize due to a lack of real world robotics data. AutoRT is a step towards addressing some of these key challenges by enabling real robot data collection while having the potential to improve state of the art robot learning algorithms, which goes beyond capabilities shown in simulated environments.
>
> “This paper lacks of technical novelty. This is a simple extension of [1, 2] to real-world robotic scenarios. I wonder if there is unique technical contribution over these existing works to deal with real-world robotic scenarios.”
> The main difference between [1,2] and our work is that AutoRT is trying to solve a different problem  –  we are trying to collect open world robotic data as opposed to learning skills in Minecraft. To this end, a significant part of the paper talks about the quality of data collected (section 5.1) and these experiments are not present in either [1,2].
> We propose a method where robot agents can ask for human help (teleoperation) or collect autonomous policies, which is not a factor in either mentioned work. This lets us scale robot deployment, where 1 human can supervise 3-5 robots in open world mobile manipulation settings due to the fact that autonomy is shared with LLMs( we have edited the paper to emphasize this). We also show how robots can ask for human help (teleoperation) or collect with autonomous policies, leading to a degree of semi autonomous operation and the ability to incorporate rich human feedback into these systems. The papers mentioned are purely autonomous and do not demonstrate sharing autonomy with humans. In fact our findings show that teleoperated data is the most diverse, and therefore, we believe that future robot data collection systems really need to incorporate many manners of collect.
> Additionally, unlike systems that work in simulation, systems for real world robotics need to keep safety at the core of its design. We introduce a robot constitution inspired from Asimov's laws, and we also show that this improves safety almost 3x. The focus and experiments on safety or attempts to align robots with human values is not part of either paper mentioned. (we have edited the paper to emphasize the contribution of real world world alignment)
>
> There is clear interest in the community for improving real-world data collection for manipulation, like ORL (https://sites.google.com/corp/view/real-orl), RoboAgent (https://robopen.github.io/), and Vision-Based Manipulators Need to Also See from Their Hands (ICLR 2022 oral, https://openreview.net/forum?id=RJkAHKp7kNZ), and we believe AutoRT builds upon these works and making progress towards addressing challenges that appear in real-world scenarios.
>
> “In table 1, the collect policies other than teleoperation have very low success rate which means the collected data might not be useful for action learning. Basically, this method sounds like it is scalable but in a low quality. Can we learn meaningful actions from these data other than visual generalization ability?”
> We would like to emphasize that AutoRT is a system for orchestrating robots, proposing tasks, and running a diverse suite of collect policies. It is not specific to the environment used or policies used for data collection. As mentioned in the reply to F9MU, the collect policies are primarily from prior work. Our contribution is in defining a way to orchestrate that prior work.
>
> The Picking (Height Generalization) evaluation task from Section 5.4 was deliberately picked as a task whose actions did not appear in teleoperated data (most teleop demos were either not on picking, or picking from a narrow range of heights). The improvement there suggests that meaningful actions can be learned from non-teleoperated data.

---

> ### Author Response · Authors · 2023-11-17
> **Rebuttal response to reviewer mkqw part2**
>
> “In Figure 3, it seems that teleoperated data has diverse action policies. However, when looking at the other data, excluding teleoperated data, they do not seem to have diverse action policies, especially considering that there are over 20 times more episodes with scripted policies. It seems that AutoRT does not explore sufficiently. Is there a way to encourage more action diversity?”
> The reviewer brings up a very interesting point about “action diversity” being an important factor in data collection. We plan for the future iterations of AutoRT to consider “action diversity” in their optimization. However, we would like to note that there is an intricate tradeoff between action diversity and safe exploration and the latter is a challenging research question. This trade off is one of many real world specific challenges that are not seen in sim. Considering the robot constitution in Section 4.4 (Affordance), we already needed to reject many generated tasks to get robot behavior within acceptable limits. Encouraging more action diversity while staying safe is an interesting line of study, but we leave it for future research.
>
>
> “Does LLM ever propose tasks that require skills not within the robot's scripted policy or teleoperation policy? If so, how does the robot behave in such cases? If such cases do not exist, how does the LLM restrict task proposals to the skills the robot possesses?”
> Most of the tasks are attemptable by the teleop policy. In the case where the LLM does propose one that cannot be done, this is usually rejected by the affordance LLM, which is prompted to reject tasks based on our description of the robot's limitations (i.e. needs 2 hands, object too heavy) and the policy's limitations (i.e. scripted policy can only do specific tasks). See Appendix D for the exact prompts and Table 4 for the quality of task generation.
> If both the initial generation and task filtering fails, the behavior depends on the policy used. For teleoperation, a human simply rejects the impossible task. For the scripted policy, we designed it to exit early instead of attempting manipulation. For RT-2, the policy will make its best attempt at the instruction. As mentioned in Section 5.3, we took a sample of 64 scenes, analyzed the failures, and found all failures in the sample were for teleoperation, rejected by the human, and not demonstrated. Our prompting for non-teleoperated policies is more conservative to make human supervision easier.
>
> “I'm curious about how robots automatically perform a reset after completing a task. Is there a predefined reset policy, or is the reset also part of the task itself? I'm curious because there seems to be not much information on the reset process in the paper.”
> In open world collect, we aimed for "no reset" as much as possible. Once a certain task is completed, the robot will re-propose tasks and the whole collect process is repeated once again. Given a large environment, it can take a while for the robot to "use up" all the opportunities to practice manipulation, since it can drive to a new scene between tasks. The battery on the mobile manipulators usually lasted 2 hours, and in addition to the robot moving due to object sampling, they were also moved for charging and after the end of the operators’ shifts.
> Manual changes to the environment could be considered as environment resets, since AutoRTs task proposals are linked to detected objects. This happened when objects were changed/redistributed by human operators, usually daily. We've updated the paper with details on this (see Appendix A).

---

> > ### Comment · Reviewer_mKqw · 2023-11-21
> > **Official Comment by Reviewer mKqw**
> >
> > Thank the authors for your kind explanations. Thanks to your explanations, most of the concerns I had have been addressed. Also your clarifications helped me understand better about the contributions of the paper and I will consider them in my final evaluation as well. Finally, I'm curious if the authors have any plans to publicly share the collected data or the code, so that others can reproduce it and conduct additional experiments.

---

> ### Author Response · Authors · 2023-11-22
> **Data and code release**
>
> We hope to release the surrounding code and collected data, but since a lot of collection happened in environments frequented by humans, we will need to spend time to ensure we can do so without breaking privacy concerns.

---

### Official Review · Reviewer_jtyJ · 2023-10-30

**Soundness:** 3 good
**Presentation:** 3 good
**Contribution:** 2 fair
**Rating:** 6
**Confidence:** 4

**Summary:**

This work proposes AutoRT, a system that uses VLM and LLM to guide diverse real-world manipulation data collection with a fleet of robots. The system proposes tasks given the scene at hand, and resort to either existing policy or human teleoperators to collect  manipulation data.
The collected data is shown to be more diverse than prior works, and lead to policy performance improvement.

**Strengths:**

* The whole system requires a lot of effort to get running
* the process of using LLM to generate diverse human-teleoperated task is interesting
* many components are interesting, presenting value to the community: robot constitution, task generation, self-reflection
* the collected data seems diverse and would be useful

**Weaknesses:**

I like the general direction, and this paper makes a first important step towards more scalable real-world data collection. I do have a somewhat philosophical question: the introduction aims to solve the problem of "lacking understanding of physical world", but at the end it primarily resort to human teleoprators to collect policies. Indeed, using LLMs to generate more diverse tasks certainly eases the cost of collecting real-world data, compared to alternatives that solely rely on human, but the stage requiring the most human effort - actual teleoperation - still remains.

Also, the idea of using llm to generate diverse tasks is not a completely new thing. I believe it was firstly proposed in [1], which talks about very similar idea of generating diverse tasks (in simulation though), and is not cited.

[1] Towards A Foundation Model for Generalist Robots: Diverse Skill Learning at Scale via Automated Task and Scene Generation (arXiv preprint arXiv:2305.10455)

**Questions:**

In the paper it says "For each environment, this map is generated once, then copied to all robots collecting in the space and loaded from cache to save time in future episodes." However, during the course of data collection, the objects and componenets in the scene will inevitably be changed. Does the map gets updated?

---

> ### Author Response · Authors · 2023-11-17
> **Rebuttal response to reviewer jtyJ**
>
> Thanks for reviewing the paper. The main weaknesses raised were that AutoRT still relies on human effort, and that using LLMs to generate diverse tasks is not completely novel. We agree that AutoRT at the moment relies on human effort, but we believe this is inevitable and AutoRT provides a promising path to reducing human effort. In addition, we would like to emphasize that AutoRT’s contributions go beyond LLM task generation. Specifically, this includes   the robot constitution for task-filtering, deciding whether to use policies that require human assistance or not, and scoring the diversity of collected episodes. These are properties that are essential when collecting data in real-world robotics settings beyond simulation environments.
>
> Questions:
>
> “I like the general direction, and this paper makes a first important step towards more scalable real-world data collection. I do have a somewhat philosophical question: the introduction aims to solve the problem of "lacking understanding of physical world", but at the end it primarily resort to human teleoprators to collect policies. Indeed, using LLMs to generate more diverse tasks certainly eases the cost of collecting real-world data, compared to alternatives that solely rely on human, but the stage requiring the most human effort - actual teleoperation - still remains.”
> It's true AutoRT still relies on some human teleoperation. This was done because our policies often failed in the more diverse scenarios AutoRT encountered, requiring teleoperation to handle them. The teleoperation needed could be reduced in 2 broad ways:
> Simplify the environment.
> Improve the robustness of the policy.
> Our goal in AutoRT was to avoid 1) as much as possible, but fundamentally AutoRT could be made more automated if the policies used were more robust. In fact, we have run AutoRT entirely autonomously in instances where we restrict the robot from moving and limit the tasks to picking tasks which we can safely run unsupervised. However, these robots did not generate diverse enough data, which led to the design choice of allowing AutoRT to use teleoperation for data collection as well. .
>
> The contribution of AutoRT is to describe an overall system where a robot can trade-off between asking for help (teleop) and running autonomously (using a policy), based on external constraints of how many people can help supervise, and further how much we can expand the robot’s capabilities using the collected data. Our hypothesis is that diversity of data plays an important role in enhancing robot capabilities, and thus we would like to effectively tradeoff between teleop and autonomous data collection.
>
> “Also, the idea of using llm to generate diverse tasks is not a completely new thing. I believe it was firstly proposed in [1], which talks about very similar idea of generating diverse tasks (in simulation though), and is not cited.
> [1] Towards A Foundation Model for Generalist Robots: Diverse Skill Learning at Scale via Automated Task and Scene Generation (arXiv preprint arXiv:2305.10455)”
>
> Thank you for this reference. We've added a citation to this work.
>
> The contribution of AutoRT, however, is not just about generating tasks via LLMs, it is the overall system for combining LLM generations with real robot execution. Outside of task generation, this includes the affordance to switch between teleop and non-teleop options, diversity scoring of the generated data, and the robot constitution to self-critique whether generated tasks are safe and within the robot's abilities. The last is especially important to real-world robot scenarios, where errors could lead to robot or environment damage.
>
> “In the paper it says "For each environment, this map is generated once, then copied to all robots collecting in the space and loaded from cache to save time in future episodes." However, during the course of data collection, the objects and componenets in the scene will inevitably be changed. Does the map gets updated?”
> The map does get updated every few hours. Importantly, the map is only used to pick navigation points. Whenever we do task proposal or manipulation, we do so based on whatever the robot's camera currently sees. Objects like desks, tables, etc. do not move during data collection, which reduces how much drift affects data collection.

---

> > ### Comment · Reviewer_jtyJ · 2023-11-22
> > **Reviewer Response**
> >
> > Thank you for your detailed reply. Some of my concerns are partially addresses. I will keep my current score.

---

### Official Review · Reviewer_F9Mu · 2023-11-01

**Soundness:** 3 good
**Presentation:** 4 excellent
**Contribution:** 2 fair
**Rating:** 5
**Confidence:** 5

**Summary:**

This paper is about using a combination of LLMs and human teleoperators to collect a large amount of diverse training data where the number of operators is less than the number of robots. The system first builds a semantic map of the environment using an open vocabulary detector + slam. It then uses the map and the list of visible objects to prompt a LLM to come up with a series of tasks. It then uses another LLM to triage the tasks between human operator, scripted pick and place, deploying rt2, or rejecting the tasks because they are too dangerous or kinematically infeasible (requiring 2-hands). The high level story of the paper makes sense but there are some questions on the experiment's side that need to be answered.

**Strengths:**

- The idea of scaling up data collection from limited human bandwidth is really good.
- Leveraging LLMs for filtering, guardrailing, and proposing tasks is also another great plus.
- Large scale data collection with  multiple robot is also commendable.

**Weaknesses:**

- The success rates are surprisingly low for both RT-1 and the one co-trained with Auto-RT. The gains are not that impressive to justify the pipeline.
- What would happen if a task is deemed solvable by RT-2 and then RT-2 cannot actually solve it?
- Does Auto-RT leverage the failure data as well?
- According to Table 1, the majority of non-teleoperated data is not successful. I would like to see a comparison with AutoRT data but only using the teleoperated portion.
- There is a confusion in the paper. Earlier RT2 is mentioned everywhere but all the tables use RT-1 for quantitative numbers. I thought RT-1 is inferior to RT-2. Why not provide quantitative numbers for RT-2 to see the gains of training on Auto-RT data are also observed there as well? I would like to see an experiment on that.
- More details are needed for the scripted policy. It seems to be dependent on object pose which is possible to extract from the cached map in the beginning but it does not say anything about the orientation of the object. Any object where the orientation of the object is important would cause failure with this policy. The only exception that I can think of where the orientation does not really matter is to pick up a deformable object such as stuffed toy, sponge, etc or symmetric objects such as bowl/plate.

**Questions:**

See weaknesses above.

---

> ### Author Response · Authors · 2023-11-17
> **Rebuttal response to reviewer F9Mu part 1**
>
> Thank you for taking the time to review the paper. AutoRT aims to be a system that is agnostic to the specifics of the policies used, such that it can deploy a wide variety of collection policies to collect data that may be useful. While we agree with F9MU's comments about the weaknesses of current policies (e.g., success rates of RT1/RT2, robustness of scripted picking policies), we would emphasize that our contribution is not in proposing better policies -- indeed, we use off the shelf prior methods for this-- but in proposing a framework that can utilize and orchestrate a variety of collection policies to enable collecting diverse data that could then be leveraged by a variety of downstream learning methods. As such, the limitations of “specific collection and learning methods, while relevant for the overall effectiveness of our system, are largely orthogonal to our contribution.
> We address individual concerns in detail below:
>
> Questions:
>
> “The success rates are surprisingly low for both RT-1 and the one co-trained with Auto-RT. The gains are not that impressive to justify the pipeline.”
> The base success of RT-1 is in fact low in our new settings. This is because Auto-RT places the robot in diverse scenarios that are significantly out of distribution from settings where the original RT-1 policies were trained. Specifically, the differing height of manipulation surfaces leads to some of the low performance we observe in the base model. However, we would like to emphasize that the performance of this base model in these novel scenarios are in fact not relevant to the AutoRT’s contribution and RT-1 is just an example of a policy one could use to initialize the learning via data collected via AutoRT.
> As for why the performance gain of RT-1 is modest, we suspect the AutoRT distribution is "flatter". It has many more tasks, and fewer episodes per task. Additionally the episodes for each task are significantly more open-world and diverse than prior robotics data, as shown in Section 5.1: Diversity Scoring. Learning quickly from such data is challenging. We posit this potentially calls for a larger scale data collection via AutoRT to observe policy improvements.
>
> As mentioned in the paper (Section 5.4), we would like to emphasize that the primary focus of AutoRT is collecting more diverse real robot data. The co-training improvement is only to show the data can be useful for training an existing method (RT-1), and  the key contribution is on an effective data collection approach, rather than learning efficiency.
>
> “What would happen if a task is deemed solvable by RT-2 and then RT-2 cannot actually solve it?”
> The episode ends in failure and is not reattempted. We experimented with multiple success detection methods, and found their precision was too low in real-world scenarios for them to be useful. In such scenarios, we rely on human labelers to provide data, which can later be used during policy improvement.
>
> “Does Auto-RT leverage the failure data as well?”
> The current learning policies of AutoRT are based on imitation learning (BC), and thus do not leverage failure data. However, one could rely on other learning algorithms that can more effectively learn from both positive and negative examples, and we hope to study this in the future.
>
> “According to Table 1, the majority of non-teleoperated data is not successful. I would like to see a comparison with AutoRT data but only using the teleoperated portion.”
> While in some settings, the non-teleoperated data seem to be unsuccessful, we would like to emphasize that there are in fact a number of settings, where the non-teleoperated data leads to policy improvements. Specifically, "picking" (height generalization) is mostly co-finetuned on data from the scripted policy, i.e., it was not very common in teleoperated data. The improvement there indicates the model can in fact improve from non-teleoperated successes.
> We are currently running an experiment using only the teleoperated portion as requested by reviewer, and will update the paper/replies once the results become available.
>
>
> “There is a confusion in the paper. Earlier RT2 is mentioned everywhere but all the tables use RT-1 for quantitative numbers. I thought RT-1 is inferior to RT-2. Why not provide quantitative numbers for RT-2 to see the gains of training on Auto-RT data are also observed there as well? I would like to see an experiment on that.”
> We have used RT-1 mainly due to practical reasons such as experiment and iteration time. While RT-2 is a more powerful model, its training can be computationally expensive (taking 2+ weeks). We have edited the paper to clarify this point.

---

> ### Author Response · Authors · 2023-11-17
> **Rebuttal response to reviewer F9Mu part 2**
>
> “More details are needed for the scripted policy. It seems to be dependent on object pose which is possible to extract from the cached map in the beginning but it does not say anything about the orientation of the object. Any object where the orientation of the object is important would cause failure with this policy. The only exception that I can think of where the orientation does not really matter is to pick up a deformable object such as stuffed toy, sponge, etc or symmetric objects such as bowl/plate.”
> We apologize if this was not clear, we have now  included the pseudocode of the scripted policy in the Appendix for more details. This policy in fact does ignore the orientation of the object. We did not spend too much time optimizing the scripted pick policy, because ultimately it is the learned policy that needs to be robust to orientation of object, and this policy has access to more data streams than the scripted pick. The goal of the scripted pick policy here is to provide episodes that the learning policy can take advantage of while leading to some level of success in many settings we encountered.

---

> ### Author Response · Authors · 2023-11-21
> **Experimental results removing non-teleoperated data**
>
> We've retrained and re-evaluated a model that was co-finetuned on only the teleop part of AutoRT data.
>
> |      | Picking (Height Generalization) | Wiping |
> | ----------- | ----------- | ----------- |
> | RT-1      | 0/24 = 0%       | 1/10 = 10% |
> | Co-fine-tuned on all AutoRT data   | 3/24 = 12.5%       | 3/10 = 30% |
> | Co-fine-tuned on only teleoperated AutoRT data | 0/24 = 0% | 2/10 = 20% |
>
> We see that non-teleoperated data was required to achieve non-zero success for height generalization and slightly helped on the wiping task.

---

### Official Review · Reviewer_P3WR · 2023-11-01

**Soundness:** 2 fair
**Presentation:** 3 good
**Contribution:** 2 fair
**Rating:** 5
**Confidence:** 4

**Summary:**

The paper discusses the challenges of training foundation models for robots due to a lack of real-world data. It introduces AutoRT, a system that uses existing foundation models to enable operational robots to work in new environments with minimal human intervention. AutoRT relies on vision-language models for scene understanding and large language models to generate diverse and innovative instructions for robots. By leveraging foundation model knowledge, AutoRT enhances data collection for robot learning while considering autonomy and safety. The system successfully deploys instructions to over 20 robots in various locations, resulting in 77k real robot episodes. Experimental results indicate that AutoRT's use of large language models leads to diverse data and robot behaviors that align with human preferences.

**Strengths:**

-The paper was well written and flows smoothly.
-The idea of a robot orchestrator is interesting, I like the idea of selecting an optimal collect policy from a suite of policies based on the generated task.
-Experiments were run to assess language and visual diversity of the dataset, and results show that the dataset has higher diversity compared to existing approaches.

**Weaknesses:**

1.The collect policies used are limited, consisting of only teleoperation, a scripted pick policy and RT2. The authors mentioned that RT2 had frequent failures, and hence decided to run RT-2 less frequently. This means that the two dominant policies were teleoperation, and scripted pick policy, both of which requires manual labor / lacks action diversity. Because of this, the execution component is not truly automated.

2.The paper emphasized the scale of the collected data, in that 53 robots were used to collect 77000 new episodes. However, while the scale of the data is big, the authors did not mention the possibility of open-sourcing the data. Further, the physical embodiment of the robot is not available to the research community.
In AutoRT Operational Details, the authors mentioned that AutoRT was run using stationary robots that skipped navigation and only conducted manipulation. In these scenarios, would the tasks which robots generate be dependent on the objects which are exposed to the robot? Can the authors provide some explanation on how the scenes for such robots were set up to ensure practicality while reducing the inherent bias that could be introduced into tasks generated by humans manually selecting the objects to set up the environment?

3.I would also suggest the author look into if there is a need for a real-robot for the data collection process as there are a few works that look into self-generating tasks or ever collecting demonstrations for a real robot without the need for a flight of robot. Examples:
-Huang, Wenlong, Chen Wang, Ruohan Zhang, Yunzhu Li, Jiajun Wu, and Li Fei-Fei. "Voxposer: Composable 3d value maps for robotic manipulation with language models." arXiv preprint arXiv:2307.05973 (2023).
-Wang, Lirui, Yiyang Ling, Zhecheng Yuan, Mohit Shridhar, Chen Bao, Yuzhe Qin, Bailin Wang, Huazhe Xu, and Xiaolong Wang. "GenSim: Generating Robotic Simulation Tasks via Large Language Models." arXiv preprint arXiv:2310.01361 (2023).
-Duan, Jiafei, Yi Ru Wang, Mohit Shridhar, Dieter Fox, and Ranjay Krishna. "AR2-D2: Training a Robot Without a Robot." arXiv preprint arXiv:2306.13818 (2023).

4. The authors specify a robot constitution for filtering tasks, where the embodiment rules involve an understanding of the physical attributes of objects (i.e. heaviness). However, it is uncertain whether the language models are suitable for embodiment style questions (i.e. weight, composition reasoning). Perhaps the authors should look into literature which studies the capability of language models and vision language models in doing physical reasoning such as:
1.Wang, Yi Ru, Jiafei Duan, Dieter Fox, and Siddhartha Srinivasa. "NEWTON: Are Large Language Models Capable of Physical Reasoning?." arXiv preprint arXiv:2310.07018 (2023).
2.Gao, Jensen, Bidipta Sarkar, Fei Xia, Ted Xiao, Jiajun Wu, Brian Ichter, Anirudha Majumdar, and Dorsa Sadigh. "Physically Grounded Vision-Language Models for Robotic Manipulation." arXiv preprint arXiv:2309.02561 (2023).

**Questions:**

All my question is listed in the weakness, I hope the authors can address my concerns in the weakness section and make the necessary changes.

**Details Of Ethics Concerns:**

Nill

---

> ### Author Response · Authors · 2023-11-17
> **Rebuttal response for reviewer P3WR part 1**
>
> Thanks for taking the time to review our paper. We appreciate the comments that AutoRT helps with robot orchestration and leads to more diverse data. The main weaknesses mentioned are the limited collect policies, questioning if real robots are needed for data collection, and potential physical reasoning errors in the LLM.  Regarding limitations of collect policies, we would like to emphasize that our contribution in AutoRT is not in proposing better policies -- indeed, we use prior methods for this -- but in proposing a framework for using a range of collect policies based on enhancing task diversity and human supervision needs. Regarding whether real robot collection is needed, we believe one could disagree on whether real robot data collection is necessary or not but this question is unrelated to judging whether a paper that proposes an improvement in real robot data collection is of high quality. AutoRT should be judged on whether it improves real robot orchestration, and not on whether real world orchestration is worth attempting given sim methods. Lastly, regarding physical reasoning errors, although LLM reasoning of physics is far from perfect, we find the semantic reasoning of LLMs with robot constitution is a combination that is still very useful for operation.
>
> 1.The collect policies used are limited, consisting of only teleoperation, a scripted pick policy and RT2. The authors mentioned that RT2 had frequent failures, and hence decided to run RT-2 less frequently. This means that the two dominant policies were teleoperation, and scripted pick policy, both of which requires manual labor / lacks action diversity. Because of this, the execution component is not truly automated.
>
> While we have settings where AutoRT runs fully autonomously, this often limits the robot environment and tasks to ones where the policies are already mostly successful. The goal of AutoRT, however, is to increase task diversity – which can potentially enhance the learned policies as well. So AutoRT attempts to avoid limiting the environment as much as possible which naturally leads to scenarios that are harder to automate. This leads to the system that needs to trade off between a diverse set of tasks that require teleoperation and less diverse tasks that can rely on already pretrained autonomous policies.
>
> However, there is no fundamental reason that AutoRT could not be more automated if the set of collect policies was more capable. Future work can focus on creating more scalable and diverse policies, that can be deployed within the system provided by AutoRT.
>
>
> 2.The paper emphasized the scale of the collected data, in that 53 robots were used to collect 77000 new episodes. However, while the scale of the data is big, the authors did not mention the possibility of open-sourcing the data. Further, the physical embodiment of the robot is not available to the research community. In AutoRT Operational Details, the authors mentioned that AutoRT was run using stationary robots that skipped navigation and only conducted manipulation. In these scenarios, would the tasks which robots generate be dependent on the objects which are exposed to the robot? Can the authors provide some explanation on how the scenes for such robots were set up to ensure practicality while reducing the inherent bias that could be introduced into tasks generated by humans manually selecting the objects to set up the environment?
>
> Yes, generated tasks depend on the objects exposed to the robot. We've edited the paper to include more details on scene setup (in Appendix A).
>
> Some of AutoRT was run in natural environments with people. Objects in those environments were biased towards typical objects (i.e. office supplies are more common in office environments). Other parts of AutoRT were run in stationary scenes or altered environments.
> For altered environments, we gathered many (over 100) random objects, like plastic toys, soda cans, stuffed animals, and office supplies. Each day some of the objects would be scattered around the scene, with a new random sample each day. This provides a greater variety of objects for AutoRT's task generation.
>
> We also believe that the language and visual diversity scoring in AutoRT could reduce the inherent bias. Assuming that human biases lead to some data being overrepresented, that data will score worse in diversity scoring, and objects can be rearranged appropriately. Appendix E shows some examples of the human supervisor arranging objects until a higher score is reached, which leads to novel scenes like turned over recycling bins.

---

> ### Author Response · Authors · 2023-11-17
> **Rebuttal response to reviewer P3WR part 2**
>
> 3.I would also suggest the author look into if there is a need for a real-robot for the data collection process as there are a few works that look into self-generating tasks or ever collecting demonstrations for a real robot without the need for a flight of robot.
>
> The methods mentioned focus on either sim or zero-shot performance. VoxPoser zero-shots affordance maps for motion planning. GenSim uses LLMs to orchestrate sim environments. AR2-D2 generate robot demos via AR from hand tracking.
>
> The question of whether such zero-shot methods remove the need for a real robot is an interesting one, but AutoRT's focus is on real robots. There is clear interest in the community for improving real-world data collection for manipulation, like ORL (https://sites.google.com/corp/view/real-orl), RoboAgent (https://robopen.github.io/), and Vision-Based Manipulators Need to Also See from Their Hands (ICLR 2022 oral, https://openreview.net/forum?id=RJkAHKp7kNZ). It's an interesting question whether sim or zero-shot methods can remove the need for real robot data collection, but this is orthogonal to judging whether AutoRT helps manage real robot data collect.
>
> Our goal in AutoRT is not to replace simulation-driven methods, but to supplement them by making it easier to scale up and deeply understand real-world robot deployments, and study how well LLMs and VLMs work in realistic scenarios.
>
> 4. The authors specify a robot constitution for filtering tasks, where the embodiment rules involve an understanding of the physical attributes of objects (i.e. heaviness). However, it is uncertain whether the language models are suitable for embodiment style questions (i.e. weight, composition reasoning). Perhaps the authors should look into literature which studies the capability of language models and vision language models in doing physical reasoning
>
> AutoRT focuses on studying what is possible using the imperfect physical reasoning abilities of current LLMs and VLMs.
>
> In terms of semantic safety of generated tasks, we find that conceptual considerations (i.e. avoid sharp things, things achievable with 1 hand) were still helpful for deploying robots and filtering tasks. Physical considerations (i.e. heaviness) were occasionally wrong, but the existing capabilities were good enough to move forward with AutoRT. We will definitely try to improve these bottlenecks in future work.

---

> > ### Comment · Reviewer_P3WR · 2023-11-23
> > **Responses for rebuttal**
> >
> > I want to thank the author for the rebuttal, i will take all of these into my final consideration.

---

### Meta-Review · Area_Chair_hncC · 2023-12-09

**Metareview:**

**Summary**
The paper proposes using foundation models to drive fleet of robots to autonomously (or at least with minimal human supervision) to collect a diverse set of real-world robot experiences.  The work proposes a system (AutoRT) that uses a open-vocabulary object detector to determine what objects are in the scenes, and then provides the list of objects to large language models (LLMs) for generating a set of instructions for the robot to follow.  To ensure that the robots operate within safety guidelines, a robot constitution is provided via prompts to the LLMs to specify constraints on what the robot is allowed to do.  Using AutoRT, a fleet of over 20 robots deployed in 4 buildings over a period of 7 months are used to collect a dataset of 77k episodes.   The episodes are collected via teleoperation and two types of autonomous robot policies (scripted pick policy and RT-2).  Analysis of the data illustrate the diversity (in language and visual observations) of the episodes.

**Strengths**
- The paper was well written with clear details on the statistics of the collected dataset and the system
- Dataset analysis show tha the collected dataset is diverse
- The dataset of collected episodes is potentially useful and interesting

**Weaknesses**
- Some ideas in the work has been explored in prior work, albeit in simulation [jtyJ,mKqw]
- Reviewers questioned whether whether a real-robot is needed for collecting the data [P3WR]
- Limited collect policies with manual control required so the system is not really fully autonomous [P3WR,jtyJ]
- Episodes with collect policies that are not teleoperation lack diversity and have low success rate, and thus may not be very useful [mKqw]
- Some details (e.g. regarding script policy) are unclear [F9Mu]
- It is not clear whether the data can be open-sourced.  Authors [in response to mKqw] indicate that they plan to release code and data but time is required due to privacy concerns.  It is unclear whether time and effort is the only issue or there may be other concerns.

**Recommendation**
Due to the weaknesses pointed out by reviewers, the AC recommends to reject the paper.

**Justification For Why Not Higher Score:**

- Given most reviewers rate the work slightly below the acceptance level, it seems that there isn't sufficient interest in the work for presentation at ICLR.  The work may be more appropriate at a robotics conference where there would be more focus on practical deployment on real-world robots.

**Justification For Why Not Lower Score:**

N/A

---

### Decision · Program_Chairs · 2024-01-16

Reject